# Unscheduled and out-of-hours care for people in their last year of life: a retrospective cohort analysis of national datasets

Bruce Mason ,[1] Joannes Joseph Kerssens,[2] Andrew Stoddart,[3] Scott A Murray ,[1] Sébastien Moine,[1,4] Anne M Finucane ,[1,5] Kirsty Boyd[1]

¹Primary Palliative Care Research Group, Usher Institute, University of Edinburgh, Edinburgh, UK
²Electronic Data Research and Innovation Service, Public Health Scotland, Edinburgh, UK
³Edinburgh Clinical Trials Unit, Usher Institute, University of Edinburgh, Edinburgh, UK
⁴Health Education and Practices Laboratory, University of Paris 13, Bobigny, France
⁵Policy and Research, Marie Curie Hospice, Edinburgh, UK

**Correspondence to**
Professor Scott A Murray;
scott.murray@ed.ac.uk

## ABSTRACT

**Objectives** To analyse patterns of use and costs of unscheduled National Health Service (NHS) services for people in the last year of life.

**Design** Retrospective cohort analysis of national datasets with application of standard UK costings.

**Participants and setting** All people who died in Scotland in 2016 aged 18 or older (N=56 407).

**Main outcome measures** Frequency of use of the five unscheduled NHS services in the last 12 months of life by underlying cause of death, patient demographics, Continuous Unscheduled Pathways (CUPs) followed by patients during each care episode, total NHS and per-patient costs.

**Results** 53 509 patients (94.9%) had at least one contact with an unscheduled care service during their last year of life (472 360 contacts), with 34.2% in the last month of life. By linking patient contacts during each episode of care, we identified 206 841 CUPs, with 133 980 (64.8%) starting out-of-hours. People with cancer were more likely to contact the NHS telephone advice line (63%) ($\chi^2$ (4)=1004, p<0.001) or primary care out-of-hours (62%) ($\chi^2$ (4)=1924,p<0.001) and have hospital admissions (88%) ($\chi^2$ (4)=2644, p<0.001). People with organ failure (79%) contacted the ambulance service most frequently ($\chi^2$ (4)=584, p<0.001). Demographic factors associated with more unscheduled care were older age, social deprivation, living in own home and dying of cancer. People dying with organ failure formed the largest group in the cohort and had the highest NHS costs as a group. The cost of providing services in the community was estimated at 3.9% of total unscheduled care costs despite handling most out-of-hours calls.

**Conclusions** Over 90% of people used NHS unscheduled care in their last year of life. Different underlying causes of death and demographic factors impacted on initial access and subsequent pathways of care. Managing more unscheduled care episodes in the community has the potential to reduce hospital admissions and overall costs.

## INTRODUCTION

Rising demand for unscheduled care is a major burden and causes pressure on healthcare systems internationally, both in and out-of-hours. Unscheduled care is unplanned

### Strengths and limitations of this study

► Linking death certification codes with service use in the 12 months before death allowed us to analyse the impact of different illnesses on unscheduled care service use.
► We linked all five unscheduled National Health Service (NHS) services (telephone advice, primary care, ambulances, emergency department and hospital admission) into Continuous Unscheduled Pathways (CUPs) and identified common patterns of unscheduled care for people.
► This innovative, population-based method provided a broad understanding of how different demographic factors affected use of unscheduled care throughout Scotland.
► Only one of the datasets, primary care out-of-hours, contained information that could indicate whether a person was identified for palliative care.
► Costs were calculated using weighted averages and were therefore approximate.

and demand-led and free at point of access. In the UK, it includes five National Health Service (NHS) services: telephone advice, primary care services, ambulance services, emergency department (ED) and acute hospital admission. ED targets were unmet throughout the UK prior to the impact of COVID-19. Unplanned hospital admissions in the UK increased by 28% from 2010 to 2019, while elective admissions rose by 25% during the same time period.[1] Contributory factors include an ageing population with multiple health conditions, public expectations, instructions to seek urgent care for suspected strokes or heart attacks, and less support in the community at weekends and overnight.[2] Much unscheduled care is used by people in their last year of life who are known to have significant health-related suffering and unmet palliative care needs.[3]

Early integration of a palliative approach as a component of chronic disease management is strongly recommended for people with advanced illnesses in all care settings.[4 5] Palliative care is a core component of universal health coverage and its people-centred ethos and focus on quality of life and death should be considered at every opportunity.[6] Services should be designed to respond to the typical trajectories of declining health of patients with all progressive illnesses, including social and psychological factors.[7 8] Palliative care includes proactive care planning which reduces burdensome interventions of low benefit, and helps avoid some unwarranted hospital admissions. However, palliative care integration into unscheduled care services has proven particularly challenging.[9 10]

National datasets can be used to monitor and improve care. They have underutilised potential to improve end-of-life care.[11–15] In Scotland, death registry data, and activity data from hospitals, the ambulance service and out-of-hours primary care services are collected routinely. These national datasets contain a unique identifier—the Community Health Index (CHI). We set out to link these data to analyse patterns of unscheduled care services use and costs by underlying cause of death and patient demographics.

## METHODS

We linked three datasets. The National Records of Scotland (NRS) deaths dataset was used to identify all adults (aged 18+) who died in Scotland in 2016. From this dataset, we extracted underlying cause of death and usual place of residence. From the General Acute Inpatient and Day Case—Scottish Morbidity Record (SMR01), we extracted all unscheduled hospital activity for the last 12 months life for the cohort. Third, we extracted data from the Scottish Unscheduled Care Datamart (UCD) covering four unscheduled care services: the telephone advice line (NHS24), primary care out-of-hours (PCOOH), the Scottish Ambulance Service (SAS) and emergency department (ED) attendances.[16] The UCD does not include in-hours, unscheduled primary care.

The ICD codes (V.2010) for underlying cause of death extracted from the NRS dataset were classified into five groups: cancer, organ failure, frailty/progressive neurological conditions, various other causes and external causes.[17] People in the first three groups were considered potentially to have had palliative care needs during their last year of life. Details of this coding allocation had been agreed previously by an expert international panel[18] (see online supplemental table 1).

Postcodes of usual place of residence were extracted from the NRS dataset. We used the Scottish Index of Multiple Deprivation (SIMD 2016) to infer quintiles of deprivation.[19] Rurality was based on the Scottish Government Urban-Rural Classification as applied to the postcodes.[20]

In order to understand how multiple services were accessed during a single healthcare episode, we used Continuous Unscheduled Pathways (CUPs) as defined in the Scottish unscheduled care datamart. A CUP is a linked set of contacts with one or more unscheduled care services.[16] Each CUP represents a single patient journey. The frequencies of the different types of CUP were tabulated to identify key patterns. We categorised a CUP as 'out-of-hours' if it started at weekends, on public holidays or on weeknights from 18:00 to 08:00. There is no limit to the duration of each CUP so a CUP could start out-of-hours but end during the in-hours period and vice versa. Therefore, only the start date and time of the CUP was used to categorise it.

In the Unscheduled Care Dataset, each service component of a CUP is assigned a code letter:
- N=NHS 24.
- O=Primary care out-of-hours.
- S=Scottish Ambulance Service.
- E=Emergency department.
- A=Acute hospital admission.

Linking these codes in chronological order gives the pathway its name. For example, 'NSE' represents a call to NHS 24 (N), followed by an ambulance service contact (S), then an emergency department attendance (E).

### Statistical analysis

Descriptive analyses were undertaken using means and frequency tables. Service use of people with or without unscheduled care contacts were compared using $\chi^2$ tests for categorical variables. Multiple logistic regression models analysed multivariate associations between predictor variables (gender, age, marital status, deprivation quintile, cause of death, urban/rural classification and place of residence) and the odds of using a service. All analyses were conducted within the Scottish National Safe Haven by a senior analyst (JJK) after approval by a Scottish Public Benefit and Privacy Panel.

### Cost estimations

Standard UK price weighting methodology was applied to estimate the costs of each unscheduled service: see online supplemental table 2 to explain how this was calculated. We did not attribute exact pricing to different forms of inpatient admission or account for differences due to patient demographics; hence,mean population values were applied. These costs are included as broad indicators of differences in scale and should not be interpreted as exact data.

### Patient and public involvement

Representatives from Marie CurieVoices Scotland and a Royal College of General Practitioners Scottish patient group joined the steering group and contributed public–patient perspectives from their own groups throughout the project. Key stakeholders from the unscheduled services and patient group members advised the research team on parameters for analysis, choice of analyses and data interpretation. To understand decision-making and experiences of service users, we conducted focus groups

**Table 1** Number (% and cumulative %) of patients (18+) in the last year of life in Scotland (2016) by number of contacts with unscheduled care services (N=56 407)

| No of service contacts | No of patients | % Patients | Cumulative % |
|---|---|---|---|
| 0 | 2898 | 5.1 | 5.1 |
| 1 | 3983 | 7.1 | 12.2 |
| 2 | 5459 | 9.7 | 21.9 |
| 3 | 5795 | 10.3 | 32.2 |
| 4 | 5205 | 9.2 | 41.4 |
| 5 | 4650 | 8.2 | 49.6 |
| 6 | 4185 | 7.4 | 57.0 |
| 7 | 3589 | 6.4 | 63.4 |
| 8 | 3090 | 5.5 | 68.9 |
| 9 | 2715 | 4.8 | 73.7 |
| 10 | 2300 | 4.1 | 77.8 |
| 11 | 1901 | 3.4 | 81.1 |
| 12 | 1646 | 2.9 | 84.1 |
| 13 | 1318 | 2.3 | 86.4 |
| 14 | 1110 | 2.0 | 88.4 |
| 15 | 967 | 1.7 | 90.1 |
| 16 | 809 | 1.4 | 91.5 |
| 17 | 679 | 1.2 | 92.7 |
| 18 | 570 | 1.0 | 93.7 |
| 19 | 520 | 0.9 | 94.6 |
| 20+ | 3018 | 5.4 | 100 |
| Total | 56 407 | 100 | |

and interviews with patients with advanced illnesses and carers who had used unscheduled services, and with bereaved carers. A final key stakeholder meeting of professionals, policy makers and lay representatives discussed the findings and implications for service development. These data will be reported elsewhere.

## RESULTS
### Underlying causes of death in the cohort
We extracted records for 56 407 adults who died in Scotland in 2016 and linked records covering the last 12 months of life for each individual in this cohort. The number of people in each disease group was as follows: cancer 28.2%, organ failure 37.7%, frailty/progressive neurological conditions 24.9%, other diseases/various causes 4.0%,and external causes 5.3% (see online supplemental table 3).

### Use of NHS unscheduled care services
The cohort had 472 360 unscheduled care service contacts; 56 407 people (94.9% of the cohort) had at least one contact. Table 1 shows the distribution of unscheduled service use: 50.4% had six or more contacts, and the 5.4% who had 20 or more contacts accounted for 21.5%

of all contacts. All unscheduled care services were used increasingly as death approached, with 34.2% occurring in the last month of life (see online supplemental table 4). During that final month, there was a disproportionate rise in primary care out-of-hours workload.

Table 2 displays the number and percentages of people who contacted the five unscheduled care services during the last 12 months of life by cause of death and demographic factors. Place of residence at death had two categories: those living in a private residence or people living in any institution. The latter were primarily care homes, but also included prisons and hostels. Due to the large sample size, all differences in table 2 were statistically significant (except contacts with NHS24 by deprivation).

### 24-HOUR TELEPHONE ADVICE SERVICE (NHS24)
More people dying with frailty (66.0%) or cancer (63.4%) contacted this service than those dying with organ failure (56.0%) ($\chi^2$ (4, N=56 407)=1004, p<0.001). People living in institutions (or commonly the staff caring for them) were more likely to contact NHS24 than those living at home: 68.8% vs 58.0% ($\chi^2$ (1, N=56 371)=352, p<0.001). People from all deprivation quintiles were about as likely to access NHS24 ($\chi^2$ (4, N=56 251)=1.88, p=0.758).

### Primary care out-of-hours (PCOOH)
Service use was similar to NHS24, as more people dying with cancer (61.5%) or frailty (61.7%) had contact with this service compared with those with organ failure (45.6%) ($\chi^2$ (4, N=56 407)=1924, p<0.001). People living in an institution were substantially more likely to have used this service than those in a private residence: 69.3% vs 50.7% ($\chi^2$ (1, N=56 371)=1011, p<0.001). People living in the most deprived quintile were less likely to access PCOOH (46.6%) compared with those from the least deprived quintile (56.9%) ($\chi^2$ (4, N=56 251)=442, p<0.001).

### Scottish Ambulance Service
People who died from organ failure used this service (78.9%) more than those with cancer (72.8%) or frailty (67.8%) ($\chi^2$ (4, N=56 407)=584, p<0.001). People living at home were much more likely to phone the ambulance service than those in an institution (77.2% vs 55.7%) ($\chi^2$ (1, N=56 371)=1725, p<0.001). People in the most deprived quintile accessed help from the ambulance service more often (77.2%) than people in the least deprived (69.5%) ($\chi^2$ (4, N=56 251)=266, p<0.001).

### Emergency department
People who died from organ failure used this service (65.2%) more than those with cancer or frailty ($\chi^2$ (4, N=56 407)=190, p<0.001), just as they did with the ambulance service. People living in institutions were less likely to visit an emergency department than those living at home (46.7% v 65.6%) ($\chi^2$ (1, N=56 371)=1250, p<0.001). Those from the most deprived quintile were more likely

**Table 2** Number (and percentage) of people who had contacts with NHS telephone advice (NHS24), primary care out-of-hours (PCOOH), Scottish Ambulance Service (SAS), emergency department (ED) and hospital admission for patients (18+) in the last year of life in Scotland (2016) by underlying cause of death, deprivation, place of residence, urban/rural classification (N=56 407)

| | All deceased persons | NHS24 | | PCOOH | | SAS | | ED attendance | | Hospital admission | |
|---|---|---|---|---|---|---|---|---|---|---|---|
| | | % with contact | No with contact | % with contact | No with contact | % with contact | No with contact | % with attendance | No with attendance | % with admission | No with admission |
| Total* | 56 407 | 59.7 | 33 656 | 53.5 | 30 161 | 73.9 | 41 678 | 62.7 | 35 383 | 74.9 | 42 253 |
| **Cause of death** | | | | | | | | | | | |
| Cancer | 15 902 | 63.4 | 10 074 | 61.5 | 9783 | 72.8 | 11 569 | 62.0 | 9857 | 88.3 | 14 039 |
| Organ failure | 21 244 | 56.0 | 11 888 | 45.6 | 9678 | 78.9 | 16 770 | 65.2 | 13 851 | 73.2 | 15 559 |
| Frailty | 14 023 | 66.0 | 9258 | 61.7 | 8654 | 67.8 | 9509 | 58.9 | 8262 | 66.4 | 9314 |
| Various | 2271 | 56.9 | 1292 | 48.9 | 1111 | 76.4 | 1735 | 64.9 | 1474 | 74.1 | 1683 |
| External | 2967 | 38.5 | 1142 | 31.5 | 935 | 70.6 | 2095 | 65.3 | 1937 | 55.9 | 1659 |
| P value | | | <0.001 | | <0.001 | | <0.001 | | <0.001 | | <0.001 |
| **Deprivation quintile** | | | | | | | | | | | |
| Most | 13 537 | 59.8 | 8093 | 46.6 | 6307 | 77.2 | 10 449 | 67.3 | 9111 | 76.4 | 10 347 |
| 2 | 12 812 | 59.8 | 7665 | 52.5 | 6720 | 76.9 | 9850 | 66.5 | 8517 | 77.2 | 9892 |
| 3 | 11 747 | 59.2 | 6959 | 55.8 | 6558 | 72.3 | 8498 | 60.0 | 7050 | 73.9 | 8680 |
| 4 | 9840 | 59.8 | 5883 | 58.8 | 5790 | 71.5 | 7032 | 58.7 | 5775 | 72.9 | 7177 |
| Least | 8315 | 60.2 | 5002 | 56.9 | 4731 | 69.4 | 5774 | 58.4 | 4853 | 72.7 | 6044 |
| P value | | | <0.758 | | <0.001 | | <0.001 | | <0.001 | | <0.001 |
| **Place of residence** | | | | | | | | | | | |
| Home/Private | 47 813 | 58.0 | 27 751 | 50.7 | 24 221 | 77.2 | 36 891 | 65.6 | 31 366 | 79.3 | 37 907 |
| Institution | 8558 | 68.8 | 5891 | 69.3 | 5928 | 55.7 | 4770 | 46.7 | 4000 | 50.6 | 4328 |
| P value | | | <0.001 | | <0.001 | | <0.001 | | <0.001 | | <0.001 |
| **Urban/rural** | | | | | | | | | | | |
| Large urban areas | 18 750 | 60.6 | 11 359 | 49.0 | 9183 | 74.9 | 14 053 | 63.3 | 11 868 | 75.4 | 14 141 |
| Other urban areas | 20 372 | 61.4 | 12 505 | 53.9 | 10 986 | 75.0 | 15 270 | 66.8 | 13 612 | 75.6 | 15 407 |
| Accessible small towns | 5244 | 57.7 | 3026 | 55.9 | 2934 | 74.1 | 3884 | 62.9 | 3297 | 74.9 | 3926 |
| Remote small towns | 2367 | 56.3 | 1332 | 59.3 | 1404 | 71.6 | 1694 | 54.8 | 1296 | 72.7 | 1721 |
| Accessible rural | 5989 | 59.6 | 3570 | 58.9 | 3526 | 71.9 | 4304 | 59.1 | 3542 | 74.3 | 4447 |
| Remote rural | 3529 | 51.3 | 1810 | 58.7 | 2073 | 68.0 | 2398 | 47.9 | 1691 | 70.8 | 2498 |
| P value | | | <0.001 | | <0.001 | | <0.001 | | <0.001 | | <0.001 |

*The deprivation, place of residence and urban/rural categories do not add up to 56 407 due to incomplete data. Each of these categories has less than 0.3% missing data.

to attend an emergency department (67.3%) compared with people in the least deprived quintile (58.4%) ($\chi^2$ (4, N=56 251)=386, p<0.001).

## Acute hospital admission

More people who died from cancer had at least one acute hospital admission (88.3%) compared with those with organ failure (73.2%) or frailty (66.4%) ($\chi^2$ (4, N=56 407)=2644, p<0.001). People living at home were much more likely to be admitted to hospital (79.3%) than those living in an institution (50.6%) ($\chi^2$ (1, N=56 371)=3043, p<0.001). People in the most deprived quintile were admitted more often (76.4%) than people in the least deprived (.72.7%) ($\chi^2$ (4, N=56 251)=134, p<0.001).

## Patterns of use across unscheduled care services

Logistic regression modelling included the clinical, socioeconomic and location variables from table 2 plus age, gender and marital status (table 3). The OR for contacts with NHS24 was higher for women than men (adjusted OR 1.17, 95% CI 1.12 to 1.21). People aged 65–84 (adjusted OR 1.68, 95% CI 1.58 to 1.78) and those aged 85 or over (adjusted OR 1.67, 95% CI 1.56 to 1.79) had higher ORs for acute hospital admission compared with people aged 18–64. Comparing the three main groups of causes of death, the odds of people with cancer having a PCOOH contact in their last year of life was much greater than for people dying with organ failure (adjusted OR 2.08, 95% CI 1.99 to 2.17). After controlling for demographic and location variables, individuals with cancer had a higher risk of an acute admission than people with organ failure (adjusted OR 2.56, 95% CI 2.28 to 2.86). Overall, people living at home used more unscheduled care services than those in institutions. This was particularly the case for acute hospital admissions (adjusted OR 3.39, 95% CI 3.21 to 3.57).

People in the most deprived quintile tended to use the three unscheduled care services that are not community based more than those in the least deprived quintile. This was especially so for ambulance services (adjusted OR 1.42, 95% CI 1.33 to 1.51). Those living in urban areas used more of all the services except PCOOH than people from rural areas overall.

## Use of continuous unscheduled pathways (CUPs)

Linking the serial use of the unscheduled service contacts into CUPs enabled us to delineate common sequences and patterns of service use. Table 4 shows the 20 most common CUPs according to when they started; in-hours or out-of-hours. NHS24 and the ambulance service were the most common initial access points. The most common end points were PCOOH and hospital admission (figure 1). Over half the ambulance calls led to an acute hospital admission (60.0%). Conversely, 50.6% of initial calls to NHS24 were dealt with in the community by PCOOH.

## Differences in frequencies and types of CUP by start time

We differentiated between CUPs that started out-of-hours or in-hours to look at implications for improving out-of-hours care as well as unscheduled care in general. We identified 206 841 continuous unscheduled pathways, of which 133 980 (64.8%) started out-of-hours, 28.1% in-hours and 7.1% unknown (mostly due to lack of a time stamp on acute hospital admissions). Contacts with NHS24 and PCOOH were much more frequent for out-of-hours CUPs. Data on the proportion of contacts with each service that occurred during out-of-hours CUPs were as follows: NHS24 93.1%, PCOOH 94.7%, ambulance service 37.7% and emergency department 44.4%.

Table 4 shows data that allow a detailed understanding of how patients typically move through the services night and day. Only 16.7% of out-of-hours CUPs started with an ambulance call, while 73.2% started with an NHS24 or PCOOH contact. Similarly, most out-of-hours CUPs ended in primary care: 9.6% with telephone advice from NHS24% and 46.7% with PCOOH. Much fewer out-of-hours CUPs resulted in an acute hospital admission (27.5%) or emergency department attendance (8.2%). In contrast, the six most common CUPs which started in-hours comprised episodes consisting of ambulance calls, emergency department visits and acute hospital admissions, and these accounted for 74.2% of all CUPs which started in-hours. GP in-hours care is not included in the UCD so was not available.

## Costs

The mean number of contacts, per patient costs and total NHS costs for the five unscheduled care services in the last year of life are listed by underlying causes of death and deprivation status in table 5. People with organ failure formed the largest group and had the highest total NHS costs as a group due to use of ambulance and hospital services. Those with frailty incurred the least unscheduled NHS care costs, being managed more in the community. The total cost of unscheduled NHS care in Scotland for people in their last year of life was nearly £190 million, of which only 3.9% was for provision of primary care services.

The total mean per-patient costs of unscheduled care in the last year of life were greatest for those with cancer (£4083), followed by organ failure (£3429) and frailty (£2654). Unscheduled per-patient costs for people in the most deprived quintile were 18.8% higher than those from the least deprived but their PCOOH costs per capita were 30.6% lower.

## DISCUSSION
## Principal findings

We found that 94.9% of people had unscheduled care contacts during their last year of life. They had a median of five contacts, with 5.1% making 20 or more, and 34.2% of all contacts occurring during the final month of life. We identified three groups of patients by underlying cause of

 

**Table 3** Adjusted OR and 95% CIs for the probability of contacts with: NHS telephone advice (NHS24), primary care out-of-hours (PCOOH), Scottish Ambulance Service (SAS), emergency department (ED) and hospital admission (HA) for patients (18+) in their last year of their life (N=56 112)*

| | NHS24 | PCOOH | SAS | ED | HA |
|---|---|---|---|---|---|
| | OR (95% CI) | OR (95% CI) | OR (95% CI) | OR (95% CI) | OR (95% CI) |
| **Sex** | | | | | |
| Male | 1 | 1 | 1 | 1 | 1 |
| Female | 1.17 (1.12 to 1.21) | 1.14 (1.10 to 1.19) | 0.93 (0.88 to 0.96) | 0.93 (0.90 to 0.97) | 1.06 (1.02 to 1.11) |
| **Age** | | | | | |
| Age 18–64 | 1 | 1 | 1 | 1 | 1 |
| Age 65–84 | 1.22 (1.16 to 1.28) | 1.18 (1.12 to 1.24) | 1.42 (1.34 to 1.50) | 1.17 (1.11 to 1.23) | 1.68 (1.58 to 1.78) |
| Age 85+ | 1.61 (1.51 to 1.71) | 1.61 (1.52 to 1.71) | 1.43 (1.34 to 1.53) | 1.12 (1.05 to 1.19) | 1.67 (1.56 to 1.79) |
| **Marital status** | | | | | |
| Single | 1 | 1 | 1 | 1 | 1 |
| Married | 1.20 (1.14 to 1.27) | 1.24 (1.17 to 1.31) | 1.13 (1.06 to 1.20) | 1.22 (1.15 to 1.30) | 1.38 (1.29 to 1.47) |
| Widowed | 1.27 (1.20 to 1.35) | 1.28 (1.21 to 1.36) | 1.13 (1.06 to 1.21) | 1.24 (1.16 to 1.31) | 1.36 (1.27 to 1.46) |
| Divorced | 1.11 (1.04 to 1.19) | 1.12 (1.04 to 1.20) | 1.11 (1.03 to 1.20) | 1.14 (1.06 to 1.22) | 1.28 (1.19 to 1.39) |
| **Deprivation** | | | | | |
| Most deprived | 1.16 (1.09 to 1.23) | 0.82 (0.78 to 0.87) | 1.42 (1.33 to 1.51) | 1.39 (1.31 to 1.48) | 1.35 (1.26 to 1.44) |
| Q2 | 1.13 (1.06 to 1.19) | 0.94 (0.94 to 0.95) | 1.39 (1.31 to 1.49) | 1.35 (1.27 to 1.43) | 1.32 (1.23 to 1.41) |
| Q3 | 1.10 (1.04 to 1.17) | 0.96 (0.96 to 0.97) | 1.20 (1.13 to 1.28) | 1.16 (1.09 to 1.23) | 1.17 (1.09 to 1.25) |
| Q4 | 1.08 (1.01 to 1.15) | 1.05 (1.05 to 1.05) | 1.17 (1.10 to 1.25) | 1.11 (1.04 to 1.18) | 1.11 (1.04 to 1.19) |
| Least deprived | 1 | 1 | 1 | 1 | 1 |
| **Cause of death** | | | | | |
| Organ failure | 1 | 1 | 1 | 1 | 1 |
| Cancer | 1.45 (1.39 to 1.52) | 2.08 (1.99 to 2.17) | 0.67 (0.64 to 0.70) | 0.81 (0.78 to 0.85) | 2.56 (2.28 to 2.86) |
| Frailty | 1.27 (1.21 to 1.33) | 1.44 (1.38 to 1.51) | 0.71 (0.64 to 0.79) | 0.77 (0.69 to 0.85) | 0.93 (0.88 to 0.97) |
| **Urban/rural** | | | | | |
| Large urban areas | 1.53 (1.42 to 1.65) | 0.72 (0.67 to 0.78) | 1.47 (1.35 to 1.60) | 1.92 (1.77 to 2.07) | 1.34 (1.23 to 1.46) |
| Other urban areas | 1.55 (1.44 to 1.67) | 0.88 (0.84 to 0.91) | 1.43 (1.32 to 1.55) | 2.24 (2.08 to 2.42) | 1.35 (1.24 to 1.47) |
| Accessible small towns | 1.33 (1.22 to 1.45) | 0.93 (0.91 to 0.94) | 1.38 (1.25 to 1.52) | 1.90 (1.74 to 2.08) | 1.26 (1.14 to 1.40) |
| Remote small towns | 1.22 (1.10 to 1.36) | 1.03 (1.04 to 1.02) | 1.23 (1.09 to 1.38) | 1.36 (1.22 to 1.51) | 1.16 (1.03 to 1.31) |
| Accessible rural | 1.44 (1.32 to 1.57) | 1.01 (0.99 to 1.02) | 1.22 (1.11 to 1.34) | 1.60 (1.47 to 1.75) | 1.15 (1.04 to 1.26) |
| Remote rural | 1 | 1 | 1 | 1 | 1 |
| **Place of residence** | | | | | |
| Institution | 1 | 1 | 1 | 1 | 1 |
| Home | 1.31 (1.24 to 1.39) | 1.89 (1.79 to 1.99) | 2.72 (2.58 to 2.87) | 2.33 (2.21 to 2.45) | 3.39 (3.21 to 3.57) |

Reference categories are male, aged 18–64, single, least deprived, organ failure, remote rural and institution (place of residence).
*295 patients in 'Other/Unknown' categories excluded. 'Various' and 'External' causes of death not shown in table.

death with different patterns of unscheduled care service use that were clinically and statistically significant. People with cancer had more unscheduled admissions than people with non-cancer diagnoses and the highest per-patient costs. People who died with frailty were most likely to have unscheduled care that was managed fully in the community. People with organ failure used most ambulance services, and as a group accounted for the greatest number of acute hospital admissions overall. People from the most deprived quintile used significantly less PCOOH than those from the least deprived but they accessed the other four unscheduled services more. The total cost of service use by people in the most deprived quintile was almost double that of those in the least deprived, due to the greater numbers of people dying in this group and their higher use of secondary care services. NHS24 and PCOOH services together accounted for less than 4% of total NHS unscheduled care costs.

**Table 4** Twenty most frequent Continuous Unscheduled Pathways (CUPs) by in-hours or out-of-hours start time for patients (18+) in the last year of life in Scotland in 2016 (N=56 407)

| | In-hours CUPs (n=58 157) | | | | Out-of-hours CUPs (n=133 980) | | | |
|---|---|---|---|---|---|---|---|---|
| Rank | Name | Number | % | CUM % | Name | Number | % | CUM % |
| 1 | SEA | 13 702 | 23.6 | 23.6 | NO | 30 434 | 22.7 | 22.7 |
| 2 | SA | 8639 | 14.9 | 38.5 | O | 18 609 | 13.9 | 36.6 |
| 3 | EA | 6408 | 11.0 | 49.5 | SEA | 9457 | 7.1 | 43.7 |
| 4 | S | 6014 | 10.3 | 59.8 | N | 8868 | 6.6 | 50.3 |
| 5 | E | 5085 | 8.7 | 68.5 | EA | 4923 | 3.7 | 54.0 |
| 6 | SE | 3291 | 5.7 | 74.2 | NSEA | 4866 | 3.6 | 57.6 |
| 7 | NO | 1450 | 2.5 | 76.7 | S | 4028 | 3.0 | 60.6 |
| 8 | N | 1230 | 2.1 | 78.8 | E | 3890 | 2.9 | 63.5 |
| 9 | O | 1005 | 1.7 | 80.5 | SE | 2816 | 2.1 | 65.6 |
| 10 | NSEA | 561 | 1.0 | 81.5 | NOSEA | 2743 | 2.0 | 67.6 |
| 11 | SEAS | 484 | 0.80 | 82.3 | OO | 2697 | 2.0 | 69.6 |
| 12 | SAO | 366 | 0.60 | 82.9 | NOSA | 2306 | 1.7 | 71.3 |
| 13 | SAS | 305 | 0.50 | 83.4 | NONO | 1677 | 1.3 | 72.6 |
| 14 | SEAO | 305 | 0.50 | 83.9 | NON | 1517 | 1.1 | 73.7 |
| 15 | SSEA | 285 | 0.50 | 84.4 | NSE | 1305 | 1.0 | 74.7 |
| 16 | SSA | 239 | 0.40 | 84.8 | NOO | 1162 | 0.90 | 75.6 |
| 17 | SS | 229 | 0.40 | 85.2 | NOA | 1112 | 0.80 | 76.4 |
| 18 | EAS | 221 | 0.40 | 85.6 | SA | 1042 | 0.80 | 77.2 |
| 19 | SES | 220 | 0.40 | 86.0 | NS | 948 | 0.70 | 77.9 |
| 20 | NSE | 164 | 0.30 | 86.3 | NOEA | 924 | 0.70 | 78.6 |

For example, the CUP 'SEA' represents a contact with the Scottish Ambulance Service (S), followed by an attendance at an emergency department (E), then an unscheduled hospital admission (A).

*14 704 missing time stamps.

A, acute hospital admission; E, emergency department; N, NHS24; O, primary care out-of-hours; S, Scottish Ambulance Service.

## Strengths and limitations of the study

### Strengths

This innovative approach to studying population use of interconnected, unscheduled care services provided a broad understanding of how different illnesses and demographic factors affected use of unscheduled care. Linking all five unscheduled NHS services together into patient pathways (CUPs) allowed common patterns of unscheduled care to be identified and quantified. Studying the unscheduled care service pathways of people in their last year of life throughout Scotland enabled us to analyse the unscheduled care provided for a whole population which has rising numbers of people with unidentified palliative care needs.[3] Understanding the perspectives and choices made by people and their families seeking unscheduled care is equally important and was the qualitative data component of our overall study (to be reported elsewhere).

### Limitations

The structure and scope of unscheduled care services in Scotland influenced how those services were used so comparisons with different countries and healthcare systems will need care. Lack of national data for in-hours, unscheduled primary care was limiting. Some CUPs recorded as out-of-hours may have started with an undocumented, urgent primary care contact in-hours. Timing of acute hospital admissions is not recorded so could not be separated by starting point. Costs were calculated using weighted averages and were therefore approximate. We acknowledge limitations in relying on ICD-10 recorded diagnoses. We had intended to look for evidence that patients had been identified for palliative care. Unfortunately, only the primary care out-of-hours dataset had a palliative care code or recorded access to the Scottish electronic care plan (Key Information Summary) used by primary care teams to coordinate palliative and anticipatory care planning.[21] This meant it was impossible to estimate the full extent of proactive care planning or palliative care provision by NHS unscheduled care services.

### Comparison with other studies

Most studies of unscheduled care have focused on individual services and specific diseases, notably emergency departments and patients with cancer, and have suggested that only 30%–35% of people with cancer use

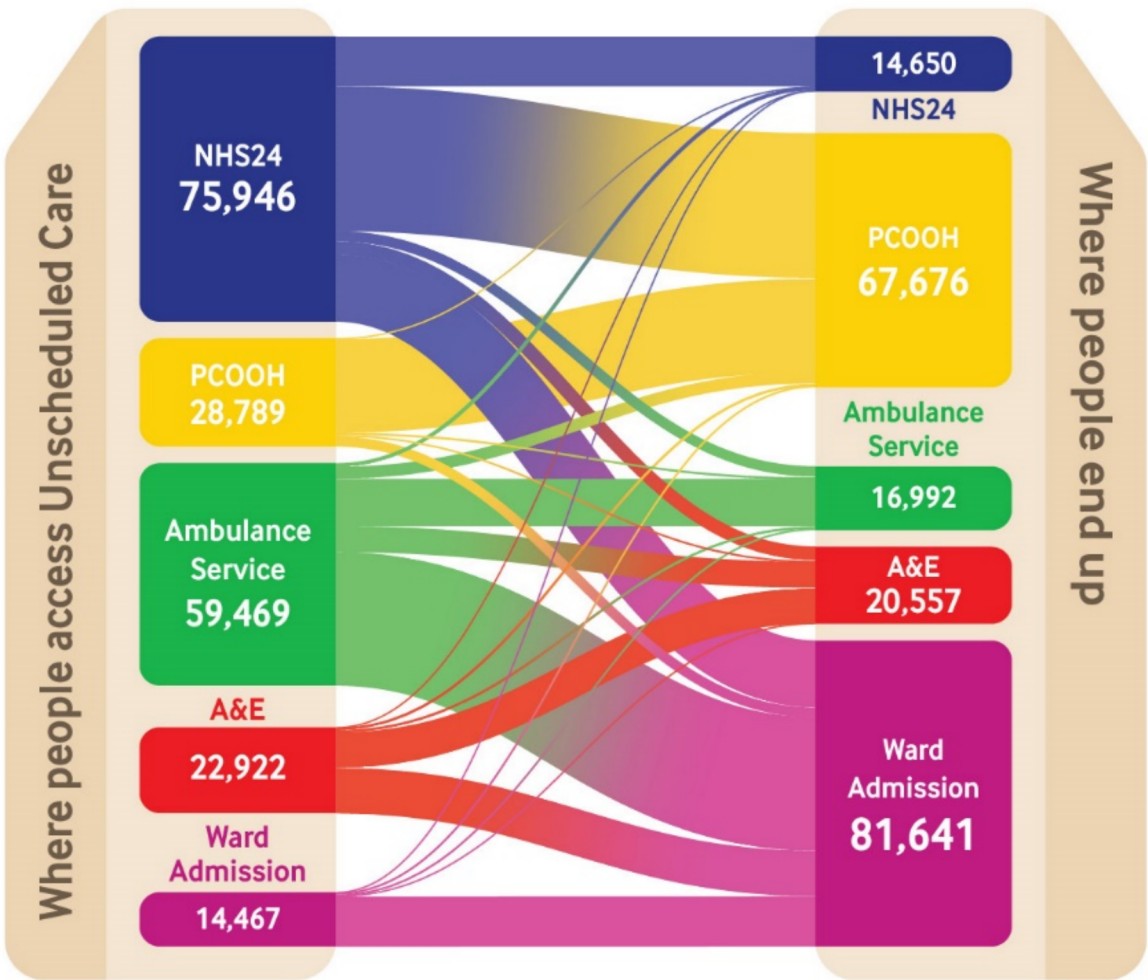

This figure shows the first and final service contacted in each Continuous Unscheduled Care Pathway (CUP). A first contact of telephone advice (NHS24) or Primary Care out-of-hours (PCOOH) was likely to be resolved in primary care. Calling an ambulance or attendance at emergency department (A&E) usually resulted in admission.

- NHS24 = telephone advice
- PCOOH= Primary Care Out of Hours
- A&E=Emergency Department

Note The figure shows 14,467 direct admissions to hospital which were likely to have been via primary care in-hours or outpatient services. Some may also be due to other direct admission routes such as the Scottish Cancer helpline for people receiving cancer treatments.

**Figure 1** Start and end points of all continuous unscheduled pathways (CUPs)

unscheduled care in the last year of life.[9 22] However, a recent study in one Scottish region reported that 78% of people dying from cancer had unscheduled care provided by emergency departments and/or PCOOH in their last year of life.[23] Our findings, integrating all five NHS unscheduled care services, found that 94.9% of people who died in Scotland received unscheduled care in their last year of life. By using population data along with specific service use data, we have highlighted the extent and diversity of unscheduled care pathways. Our logistic regression models identified differences in unscheduled care by deprivation quintile for each of the three main illness groups, not just for the population with cancer, and correlates with other evidence around the relationship between socioeconomic status and use of emergency services.[9 11 13]

**Table 5** Mean number of contacts and costs of NHS telephone advice (NHS24), primary care out-of-hours (PCOOH), Scottish Ambulance Service (SAS), emergency department (ED) and hospital admissions for patients (18+) in the last year of life in Scotland (2016) by cause of death and deprivation status (=56 407)

| | NHS24 | | | PCOOH | | | SAS | | | Emergency department | | | Hospital admissions | | | All contacts | | |
|---|---|---|---|---|---|---|---|---|---|---|---|---|---|---|---|---|---|---|
| | Mean number of contacts | Total costs (×1000) | Costs per patient | Mean number of contacts | Total costs (×1000) | Costs per patient | Mean number of calls | Total costs (×1000) | Costs per patient | Mean number of attendances | Total costs (×1000) | Costs per patient | Mean number of admissions | Total costs (×1000) | Costs per patient | Total costs to NHS (×1000) | Total cost to NHS per patient |
| Total | 1.4 | £2944 | £52 | 1.3 | £4471 | £79 | 1.7 | £22337 | £396 | 1.3 | £10205 | £181 | 1.7 | £149967 | £2659 | £189923 | £3367 |
| **Cause of death** | | | | | | | | | | | | | | | | | |
| Cancer | 1.4 | £836 | £53 | 1.6 | £1502 | £94 | 1.6 | £5807 | £365 | 1.3 | £2765 | £174 | 2.1 | £54022 | £3397 | £64933 | £4083 |
| Organ failure | 1.3 | £1047 | £49 | 1.1 | £1370 | £64 | 2.0 | £9525 | £448 | 1.4 | £4173 | £196 | 1.7 | £56730 | £2670 | £72845 | £3429 |
| Frailty | 1.7 | £842 | £60 | 1.6 | £1315 | £94 | 1.5 | £4892 | £349 | 1.1 | £2165 | £154 | 1.3 | £28003 | £1997 | £37218 | £2654 |
| **Deprivation** | | | | | | | | | | | | | | | | | |
| Most deprived | 1.5 | £748 | £55 | 1.1 | £794 | £59 | 1.9 | £5960 | £440 | 1.5 | £2867 | £212 | 1.8 | £38809 | £2867 | £49177 | £3633 |
| 2 | 1.5 | £678 | £53 | 1.3 | £984 | £77 | 1.9 | £5543 | £433 | 1.4 | £2559 | £200 | 1.8 | £35805 | £2795 | £45569 | £3557 |
| 3 | 1.4 | £607 | £52 | 1.5 | £1048 | £89 | 1.6 | £4390 | £374 | 1.2 | £1936 | £165 | 1.6 | £29625 | £2522 | £37606 | £3201 |
| 4 | 1.4 | £501 | £51 | 1.6 | £929 | £94 | 1.6 | £3644 | £370 | 1.2 | £1569 | £159 | 1.6 | £25050 | £2546 | £31693 | £3221 |
| Least deprived | 1.4 | £405 | £49 | 1.4 | £710 | £85 | 1.5 | £2766 | £333 | 1.1 | £1256 | £151 | 1.5 | £20349 | £2447 | £25487 | £3065 |

## Meaning of the study and implications for clinicians and policy makers

Many more people seek unscheduled and out-of-hours care in their last year of life than was recognised previously. There were common patterns associated with different underlying illnesses, deprivation status and place of residence. Knowledge of how these groups of patients respond to urgent care needs may help community and hospital services find ways to respond more effectively and potentially could reduce demand for costly services. Primary care teams, social care managers and hospital teams can identify frequent or unusual patterns of contacts with unscheduled care and use these to trigger new or updated care planning. Such care planning communicated to unscheduled care services routinely via primary care managed electronic care coordination systems has been linked with fewer hospital admissions and deaths in Scotland.[21 24] In London, the 'Coordinate my Care' system uploads and shares urgent care plans entered by primary care, ambulance and hospital services thereby reducing hospital admissions, and electronic care planning systems are evolving in other parts of England.[10 25] The UK ReSPECT process partners with patients and families to make emergency treatment and care plans that can help guide unscheduled care and reduce unwarranted admissions.[26] Key aspects are early identification of people at risk of deteriorating health, proactive care planning and a readily accessible electronic care coordination system that can be read and updated by any professional responsible for a person's care.

Improving the ability of unscheduled primary care services to manage people in the community is likely to be highly cost-effective as well as supporting people's choice to remain at home towards the end of life. Emergency departments are already employing more primary care clinicians to enhance prehospital triage and ambulance crews are providing more care at home, where appropriate, instead of transferring patients to hospital.[27] Interventions in the community by NHS 24, PCOOH and ambulance services have potential to provide high-value low-cost care.

## Further research

Unscheduled care of the whole population merits ongoing research using population-level data, encompassing all community and hospital settings both in-hours and out-of-hours. Improving the scope and quality of data collected routinely can facilitate research into the needs of people who are high service users and stand to benefit from better coordinated care.[5 6 28] Specifically, research to understand specific differences in care pathways and service use is important, such as why people with cancer have more urgent hospital admissions than others and why people with organ failure call the ambulance service relatively frequently. Our data also provide a baseline that can be used in studies to evaluate changes in the use of unscheduled services during the coronavirus pandemic, when the demand for hospital-based unscheduled care dipped sharply.

Interventions to encourage a palliative care approach in each of the five out-of-hours services, as well as care coordination throughout the unscheduled pathways are recommended.[29] An evaluation of telephone advice services including emergency social care, community nurse telephone support such as the Gold Line, specialist palliative care support lines, and from charities offering help for people with specific illnesses is indicated to scope provision of unscheduled care in the community further.[29] Research into the contribution of specialist palliative care out-of-hours, which has not been included in this study, and interventions to coordinate care between settings at the end of life are also needed.[29]

## CONCLUSIONS

The extent of unscheduled care delivered to people in their last year of life is significantly greater and more varied than reported previously. People with diverse urgent care needs are accessing these services at high levels, particularly in their final month of life. More should be done to take account of underlying illness trajectories and social determinants of health, including better public understanding of how to access the right care in timely and effective ways. Systematic approaches to care planning combined with effective recording and sharing of key information, including a palliative care code where appropriate, is vital and should be recorded in routine healthcare datasets.

**Acknowledgements** The authors acknowledge the support of National Services Scotland for their involvement in obtaining approvals, provisioning and data linkage. We also thank Bruce Guthrie and Sarah Mills for commenting on a draft of this paper.

**Collaborators** Primary Palliative Care Out-of-hours Project Steering Group: Marilyn Kendall, senior researcher; Sheonad Laidlaw, general practitioner; Emma Carduff, honorary research fellow and Marie Curie research lead; Erna Haraldsdottir, senior lecturer; Sir Lewis Ritchie, Scottish Government advisor; Sian Tucker, NHS24 national lead; Marie Fallon, professor of palliative medicine; Jeremy Keen, consultant in palliative medicine; Stella Macpherson and Lorna Moussa, patient and public involvement representatives.

**Contributors** Contributor and guarantor information: BM, SAM, SM, AF, AS and KB designed the study (along with members of the steering group acknowledged above); JJK and AS linked and analysed the data, SM allocated illness trajectories to all decedents; and AF conducted literature reviews. All contributors discussed the findings and contributed to writing the drafts and the final paper. The contributors met with the steering group regularly to design and guide the overall study and to integrate patient and carer perspectives. SAM and KB are the guarantors. The corresponding author attests that all listed authors meet authorship criteria and that no others meeting the criteria have been omitted.

**Funding** Combined grant MCRGS-07-16-37 from Marie Curie and the Chief Scientist Office Scotland.

**Competing interests** No support from any additional organisations for the submitted work; no financial relationships with any organisations that might have an interest in the submitted work in the previous three years; no other relationships or activities that could appear to have influenced the submitted work.

**Patient consent for publication** Not required.

**Ethics approval** Permission to access and link data from the Scottish Public Benefit and Privacy Panel for health and social care, ref 1516-0483 and approval from SE Scotland Research Ethics committee 2 reference 17/SS/0127 were gained.

**Provenance and peer review** Not commissioned; externally peer reviewed.

**Data availability statement** Data are available upon reasonable request. More information may be available from the corresponding author.

**ORCID iDs**
Bruce Mason http://orcid.org/0000-0002-9304-3362
Scott A Murray http://orcid.org/0000-0002-6649-9428
Anne M Finucane http://orcid.org/0000-0002-3056-059X

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
