## [Reviewer comments · BMJ Open]

ARTICLE DETAILS

TITLE (PROVISIONAL)	Unscheduled and out-of-hours care for people in their last year of life: a retrospective cohort analysis of national datasets
AUTHORS	Mason, Bruce; Kerssens, Joannes; Stoddart, Andrew; Murray, Scott; Moine, Sébastien; Finucane, Anne; Boyd, Kirsty

VERSION 1 – REVIEW

REVIEWER	Bee Wee 1. Harris Manchester College University of Oxford 2. Oxford University Hospitals NHS Foundation Trust United Kingdom
REVIEW RETURNED	16-Jul-2020

GENERAL COMMENTS	This is an excellent and timely study, well designed and executed. The methodology is well described and could be replicated in other countries. I have a few comments: 1. Page 8 - typo in 2nd line - reference to online supplementary table 3, should be table 4.2. Page 12 - lines 7-12. You state that people with organ failure made the most use of ambulance services and accounted for the greatest number of acute hospital admissions overall due to there being more deaths in this group. I think you are inferring a causal link with the greater number of deaths when it may not be the only factor that accounts for the greater number of hospital admissions. If your data does demonstrate such a link, it would be helpful to demonstrate that more explicitly.3. Page 13 - section on limitations. There should be an acknowledgement that the findings of this study may, or may not, be specific to Scotland. It is possible that different groups of patients may respond to their urgent care needs differently depending on the different design of acute, community and unscheduled care services available to them in different countries and environments. Otherwise I think this is an interesting, helpful and clearly written publication.
---

REVIEWER	Joanna Davies King's College London, UK
REVIEW RETURNED	04-Aug-2020

GENERAL COMMENTS	Comments to the authors: Thank you for this paper that links up several sources of routine data to offer a comprehensive picture of the use of unscheduled care in the last year of life. It's a great example of what is possible with linked data. Figure 1 is a really useful graphic for summarising
---

a large amount of data. Most of my comments are minor with three major points I ask you to consider.

Major comments:

1. Did you consider a regression of cost on your exposure variables? It would potentially be more informative to predict cost (either total or patient or both) using adjusted regression and then report the associations. Is there a reason you didn't do this?
2. Please add p values to table 2. Although this will introduce a lot of statistical tests I think it will help the reader to identify the significant differences – you can use a bonferoni adjustment to account for the multiple tests
3. In table 3, please change the reference categories so they are consistent across the models. It doesn't matter if some OR are above 1 and some below – it is more intuitive to have consistent ref categories than to seek a consistent direction.

Minor comments:

Abstract

The abstract could do with rewording, there is a lot of information, choose the most important and be more specific in the results. For example, say which groups used unscheduled care more/cost most. Your explanation of CUP could be both clearer and more succinct. Please say what the 5 unscheduled services are.

Introduction

- You say integration of pall care into unscheduled care is particularly challenging – can you expand and say why?
- In the aim (line 33) specify what element of cost you looked at.

Methods

- What is the Scottish morbidity record? – if it's important to mention it can you give a bit more information please?
- More information here on the cost estimations please, explain what you looked at and how it was calculated.

Results

- The results could do with tightening up. In particular, please always give the statistical results that you discuss in the text – in some cases the % is missing. Also add the p values (in reference to my major comment above). When you give the % for the most deprived, also give it for the least deprived.
- “older people aged 85 or over had slightly lower ratios, likely due to care home residence (adjusted OR 1.67, 95% CI = 1.56-1.79).” – are these the results for the over 85 or for care home residents? This sentence seems to conflate the two exposures. Save any explanation of the results for the discussion. Also ‘ratios’ should be ‘odds’. Similarly - “probably due to service organisation in remote, rural areas in Scotland” - would be better saved for the discussion.
- “percentages of contacts with each service that occurred during out-of-hours CUPs compared with in-hours CUPs” – loose the ‘compared with...’
- Table 2 – im not sure what the foot note means? Are you referring to missing data? If so please report the missing data in the tables, or if this is not possible for presentation reasons include a detailed footnote summarising how much missing for each variable.
- In all of the tables, please include footnotes for any abbreviations like SAS to help the reader understand
- Table 3 – is age 18-64 the ref cat? – you have left out the 1

	Discussion  • “People who died with frailty, especially those who lived in institutions” – this suggests you did a subgroup analysis of people in institutions which you didn’t. • You could reference this study Walsh B, Laudicella M. Disparities In Cancer Care And Costs At The End Of Life: Evidence From England's National Health Service. Health Aff (Millwood). 2017;36(7):1218-1226.
--	--

REVIEWER	Danielle Ni Chroinin Department of Geriatric Medicine, Liverpool Hospital, and South Western Sydney Clinical School, UNSW Sydney. I have co-authored a paper entitle "Health-services utilisation amongst older persons during the last year of life: a population-based study" (BMC Geriatr 2018). No other conflicts to declare.
-----------------	--

REVIEW RETURNED	10-Aug-2020
-------------

GENERAL COMMENTS	Unscheduled and out-of-hours care for people in their last year of life: a population analysis in a national health care system Thank you for the opportunity to review this paper, which I believe will be of interest to the readers of BMJ Open, although it would benefit from minor revisions. The authors report the utilisation of unscheduled care- accessed through 5 different care services- by adults dying in Scotland over a 12 month period. This represents an innovative use of national datasets, and includes graphic representation of the patient journey throughout an episode of unscheduled care. The study is strengthened by its large numbers, Scotland-wide scope, and the analysis of healthcare use according to patient characteristics such as age, sex, residence, rurality and deprivation (index). Figure 1 gave a helpful pictorial overview of the flow of ‘CUP’ episodes. The authors also highlight the importance to explore patient experiences and priorities, and note that these will be reported elsewhere. Comments/queries: 1) Within the Introduction, the authors highlight the importance of palliative care and its role in advanced illness and potential to minimise low-benefit interventions towards end of life. Unfortunately, the lack of comprehensive palliative care information means that palliative care could only be identified within primary care out-of-hours services (PCOOH), and within these, by a combination of palliative care ‘Read codes’ and inferentially through the use of KIS. As advance/anticipatory care planning (ACP) need not (should not) be limited to those who are approaching end-of-life, the authors have perhaps made a bit of a jump in equating KIS with ‘identification for palliative care’. (For example, Tapsfield et al, BMJ Supportive and Palliative Care,
---

2016, noted that individual GP practices identified up to 92% of patients as being appropriate for KIS.) While using KIS as a marker of possibly having advance care planning input may not be unreasonable as a pragmatic approach, the authors should wish to expand on this in the Discussion. Furthermore, while the authors flag the lack of routinely collected palliative care code is a limitation for this study, but perhaps more importantly, it identifies a deficit in data collection within many (not just Scottish) national/healthcare databases, and the authors may wish to highlight this.

2) The authors harvested important information from large available datasets. This approach has been adopted by other authors investigating use of healthcare utilisation in the last 6-12 months of life, using national or insurance databases in the United States, Australia and Europe. These have not all been confined to ED presentations or disease-specific cohorts, and the authors may wish to update the text (Intro/Discussion) to reflect this.

3) Frailty was a relatively common cause of death with increasing age, and was identified as present in almost 1 in 5 of decedents aged 80+. Frailty (not unlike palliative care) is often poorly recognised and poorly captured. As the authors have included frailty/progressive neurological conditions as one of five different disease groups, it may be appropriate to mention gaps in/limitations of relying on ICD-10 recoded diagnosis.

4) Residents of institutions (mostly nursing homes) were lower users of many unscheduled care services. This may be as a result of many factors, including higher rates of ACP (if mandated/encouraged within the residential facility), available nursing care, and specialist outreach (or in-reach) palliative care and geriatric services. Data regarding any correlations between these factors and use of unscheduled care services in this study would be helpful and of interest, but may not be available. I note later in the discussion, the authors acknowledge lack of data regarding specialist palliative care out-of-hours.

5) In this study, those from more deprived areas use unscheduled services differently, and in particular access unscheduled care more often via ambulance, and less often e.g. via PCOOH. This finding is similar to that identified by previous authors in relation to emergency healthcare use and death-in-hospital (e.g. review by Davies et al, PLOS Medicine 2019). The authors may wish to

hypothesis as to why these differences exist, with reference to the appropriate literature.

6) The authors note that “Improving the ability of unscheduled primary care services to manage people in the community is likely to be highly cost-effective”. While this is a fair generalisation, it would be helpful to know the reasons why patients are directed/diverted to ambulance/emergency/hospital services. What is the care gap? Lack of ACP? Lack of investigations e.g. pathology, scanning? Lack of equipment e.g. oxygen, hospital beds? Lack of specialist/expert opinion to assist in management? Lack of allied health availability? Lack of community palliative care? This information will help guide where resources should be invested.

In relation to Table 4:

- The authors mention in the methods and results that GP in-hours care data are not available. The relatively high apparent default to/reliance on ambulance and emergency department care in the in-hours cohort may in part be influenced by a GP possibly having reviewed the patient (in person or by phone), and advised a need for emergency presentation. The apparent discrepancy in community support use between in-hours and out-of-hours services might not be so marked if in-hours unscheduled GP visits/calls were included. This probably needs to be explicitly stated.
- A surprisingly high proportion of patients seem to have bypassed ED, going straight from ambulance to admission (per Table 4). Do the authors wish to comment on this, in particular its possible relationship to direct admission to specialist wards/care and avoidance of ED ‘harms’?
- Can I please ask how some patients accessed ‘out of hours’ primary care, seemingly ‘within hours’ and as the first port of call (i.e. not diverted e.g. from NHS24)?

Minor points:

Given the complex interplay of factors involved- not all of which may be “fix-able”, I would suggest substituting the use of the word “unwarranted” (in relation to e.g. admissions) with “potentially avoidable” or similar.

It might be reasonable to note that some of the lower ‘unscheduled healthcare’ resource use by residents of institutions might be offset by the higher day-to-day care provision.

For readers outside of Scotland, it might be helpful to summarise the NHS24 line service (which appears to be staffed by nurses, pharmacy advisers and health information advisers) in a line.

	On page 7, the statistic of 56,407 people should include either “aged ≥18 years” or use “adults”. The authors note in Supplementary material (Table 2 and following text) that they did not analyse cost by length-of-stay and procedures undertaken. It was possibly beyond the scope of this study, but these data would be interesting.
--	--

VERSION 1 – AUTHOR RESPONSE

Reviewer 1 (Bee Wee)

1. Page 8 - typo in 2nd line - reference to online supplementary table 3, should be table 4.	Thanks. Done p7.
2. Page 12 - lines 7-12. You state that people with organ failure made the most use of ambulance services and accounted for the greatest number of acute hospital admissions overall due to there being more deaths in this group. I think you are inferring a causal link with the greater number of deaths when it may not be the only factor that accounts for the greater number of hospital admissions. If your data does demonstrate such a link, it would be helpful to demonstrate that more explicitly.	Sorry we did not mean to imply this causal link so have removed the ambiguous statement We have clarified this in the abstract and under the Costs paragraph in the results p10: ‘People dying with organ failure formed the largest group in the cohort which had the highest total NHS costs due to use of hospital services.’
3. Page 13 - section on limitations. There should be an acknowledgement that the findings of this study may, or may not, be specific to Scotland. It is possible that different groups of patients may respond to their urgent care needs differently depending on the different design of acute, community and unscheduled care services available to them in different countries and environments.	Done - this caveat is inserted on p12

Reviewer 2 Reviewer Name: Joanna Davies, Institution and Country: King's College London, UK

Major	
Did you consider a regression of cost on your exposure variables? It would potentially be more informative to predict cost (either total or patient or both) using adjusted regression and then report the associations. Is there a reason you didn't do this?	We considered it but did not do so. The price weights are simple scalars. Any regression which predicts the pattern of health care utilisation on a given element would also predict the cost for that element. If you were to combine all elements, then because the Inpatient costs are so much larger, it

	would likely render the other factors moot. And since the Inpatient price weighting is the most caveated, there would be a big risk of over interpretation here.
Please add p values to table 2. Although this will introduce a lot of statistical tests I think it will help the reader to identify the significant differences – you can use a bonferoni adjustment to account for the multiple tests	Done new table 2 now clearly shows 5 new rows of p values
In table 3, please change the reference categories so they are consistent across the models. It doesn't matter if some OR are above 1 and some below – it is more intuitive to have consistent ref categories than to seek a consistent direction.	Done- we agree it is more intuitive to have consistent ref categories: see new table 3
Minor comments	
The abstract could do with rewording, there is a lot of information, choose the most important and be more specific in the results. For example, say which groups used unscheduled care more/cost most. Your explanation of CUP could be both clearer and more succinct. Please say what the 5 unscheduled services are.	Abstract reviewed and reworded. We have specified which groups used more unscheduled care and clarified findings on costs in line with comments from reviewer 1 & 2 Explanation of CUPs made clearer and shorter p 5. The 5 unscheduled care services are now listed in article summary: strengths and weaknesses on p3 as no room in the abstract
You say integration of pall care into unscheduled care is particularly challenging – can you expand and say why?	Done. We have changed the text slightly to make it clear that we are referencing an argument rather than making one. We have added a further reference to support this, ref10 p4.
In the aim (line 33) specify what element of cost you looked at.	Clarified on p 4 at end of introduction
Methods. What is the Scottish morbidity record? – if it's important to mention it can you give a bit more information please?	Updated the nomenclature. p 4
Methods. More information here on the cost estimations please, explain what you looked at and how it was calculated.	We have inserted a phrase in the text on p 7 to point to supplementary table where these details are provided.
Results	
The results could do with tightening up. In particular, please always give the statistical results that you discuss in the text – in some cases the % is missing. Also add the p values (in reference to my major comment above). When you give the % for the most deprived, also give it for the least deprived.	Done throughout the results. Statistical results and % inserted, including p values. % of least deprived also inserted top of p9

“older people aged 85 or over had slightly lower ratios, likely due to care home residence (adjusted OR 1.67, 95% CI = 1.56-1.79).” – are these the results for the over 85 or for care home residents? This sentence seems to conflate the two exposures. Save any explanation of the results for the discussion. Also ‘ratios’ should be ‘odds’. Similarly - “probably due to service organisation in remote, rural areas in Scotland” - would be better saved for the discussion.	The speculation is removed and this paragraph tightened up according to available data. Ratios changed to odds. P11 We have removed the text relating to rurality as remote-rural populations were a small proportion of the cohort.
“percentages of contacts with each service that occurred during out-of-hours CUPs compared with in-hours CUPs” – lose the ‘compared with...’	done
Table 2 – im not sure what the foot note means? Are you referring to missing data? If so please report the missing data in the tables, or if this is not possible for presentation reasons include a detailed footnote summarising how much missing for each variable.	Footnote* The deprivation, place of residence and urban/rural categories do not add up to 56,407 due to incomplete data. We have added: “Each of these large categories has less than 0.3% missing data”
In all of the tables, please include footnotes for any abbreviations like SAS to help the reader understand	Now Incorporated in titles in all tables.
Table 3 – is age 18-64 the ref cat? – you have left out the 1	Now corrected in new table 3
Discussion	
“People who died with frailty, especially those who lived in institutions” – this suggests you did a subgroup analysis of people in institutions which you didn’t.	Phrase removed as unnecessary in this paper
You could reference this study Walsh B, Laudicella M. Disparities In Cancer Care And Costs At The End Of Life: Evidence From England's National Health Service. Health Aff (Millwood). 2017;36(7):1218-1226.	Many thanks for suggestion. Now incorporated as new ref 15

Reviewer 3 Danielle Ni Chroinin

Institution and Country: Department of Geriatric Medicine, Liverpool Hospital, and South Western Sydney Clinical School, UNSW Sydney, Australia Please state any competing interests or state ‘None declared’: I have co-authored a paper entitle "Health-services utilisation amongst older persons during the last year of life: a population-based study" (BMC Geriatr 2018).

While using KIS as a marker of possibly having advance care planning input may not be unreasonable as a pragmatic approach, the authors should may wish to expand on this in the Discussion.	We did this, but on balance we decided to omit this short paragraph in methods as we do not present numerical results about identification. We do stress the following more general point however.
Furthermore, while the authors flag the lack of routinely collected palliative care code is a limitation for this study, but perhaps more importantly, it identifies a deficit in data collection within many (not just Scottish) national/healthcare databases, and the authors may wish to highlight this.	Yes a very important point, thanks. Now highlighted in conclusions on p15.
The authors harvested important information from large available datasets. This approach has been adopted by other authors investigating use of healthcare utilisation in the last 6-12 months of life, using national or insurance databases in the United States, Australia and Europe. These have not all been confined to ED presentations or disease-specific cohorts, and the authors may wish to update the text (Intro/Discussion) to reflect this.	Thanks We have added new references. Walsh 2017, Ni Croinin 2018, Asaria et al 2016, davies et al 2019 See11-14
Frailty As the authors have included frailty/progressive neurological conditions as one of five different disease groups, it may be appropriate to mention gaps in/limitations of relying on ICD-10 recoded diagnosis.	We have acknowledged this on p12 “We acknowledge limitations in relying on ICD-10 recorded diagnoses” We also know that patients in the other groups may indeed have frailty as well. Our coding has tried to capture the predominant illness trajectory
4) Residents of institutions (mostly nursing homes) were lower users of many unscheduled care services. This may be as a result of many factors, including higher rates of ACP (if mandated/encouraged within the residential facility), available nursing care, and specialist outreach (or in-reach) palliative care and geriatric services. Data regarding any correlations between these factors and use of unscheduled care services in this study would be helpful and of interest, but may not be available. I note later in the discussion, the authors acknowledge lack of data regarding specialist palliative care out-of-hours.	Thanks for sharing these insights. We will consider this as we analyse further data in the qualitative arm of this study.
5) In this study, those from more deprived areas use unscheduled services differently, and in particular access unscheduled care more often via ambulance, and less often e.g. via PCOOH. This finding is similar to that identified by previous authors in relation to emergency healthcare use and death-in-hospital (e.g.	Added sentence to end of Comparison with other studies We will be better able to hypothesise when we have analysed the qualitative data from patients and their informal and professional out-of-hours carers in the next publication.

review by Davies et al, PLOS Medicine 2019). The authors may wish to hypothesise as to why these differences exist, with reference to the appropriate literature.	
6) The authors note that "Improving the ability of unscheduled primary care services to manage people in the community is likely to be highly cost-effective". While this is a fair generalisation, it would be helpful to know the reasons why patients are directed/diverted to ambulance/emergency/hospital services. What is the care gap? Lack of ACP? Lack of investigations e.g. pathology, scanning? Lack of equipment e.g. oxygen, hospital beds? Lack of specialist/expert opinion to assist in management? Lack of allied health availability? Lack of community palliative care? This information will help guide where resources should be invested.	Again we will attempt to address these questions using the qualitative data in the next publication and make suggestions to improve the generalist OOH care for people in their last year of life (whether they know it or not!)

VERSION 2 – REVIEW

REVIEWER	Joanna Davies King's College London, UK
REVIEW RETURNED	25-Sep-2020

GENERAL COMMENTS	Thank you for addressing my earlier comments. The results are much clearer now, however, please can you report the statistical result more fully i.e. at least include the X2 results and degrees of freedom alongside the p value. Please also follow this approach in the abstract - and include in the abstract the % as well as the (X2 (df) =??, p value). See the APA style, or examples here: http://ich.vscht.cz/~svozil/lectures/vscht/2015_2016/sad/APA_style2.pdf http://users.sussex.ac.uk/~grahamh/RM1web/APA%20format%20for%20statistical%20notation%20and%20other%20things.pdf
--

REVIEWER	Danielle Ni Chroinin Department of Geriatric Medicine, Liverpool Hospital, and South Western Sydney Clinical School, UNSW Sydney. I have co-authored a paper entitled "Health-services utilisation amongst older persons during the last year of life: a population-based study" (BMC Geriatr 2018)
REVIEW RETURNED	29-Sep-2020

GENERAL COMMENTS	Thank you for the opportunity to review this revised manuscript. The authors appear have addressed the major points raised by all
---

	3 reviewers. The paper appears suitable for publication in its current guise, with a couple of very minor suggestions: Abstract: The phrasing here isn't intuitive to me: "People dying with organ failure formed the largest group in the cohort which had the highest NHS costs overall." Is this 'and had the highest NHS costs as a group'? Discussion: "People with organ failure formed the largest group which had the highest total NHS costs due to use of hospital services." Likewise. Also, their hospital-related costs are only slightly more than the group with cancer. Use of ambulance services also accounts for a chunk of the difference, and so maybe one than factor should be included.
--	--

VERSION 2 – AUTHOR RESPONSE

Reviewer 2: We have reported the statistical results more fully, including the X2 results and degrees of freedom alongside the p values throughout in the results, and in the abstract. We have also added the %s in the abstract as well as the (X2 (df) =??, p value. We have removed a few words from the abstract to keep it under 300 words

Reviewer 3: We have adopted the suggested phrasing re organ failure which is clearer: many thanks. We have done this in the abstract and later on at the end of results (although she wrote "Discussion" above, she referred to sentences in the "Costs" section at the end of results) We have now inserted "ambulance services" as suggested as they indeed are highly used by this group of patients.(page 11)

I final point is that I have tried to list the Primary Palliative Care Out-of-hours Project Steering Group as an author in step 5 where I have listed their names, but technically unable to do that. It would be good to have that group acknowledged as collaborators if possible please.

Many thanks, and we are very grateful for your willingness to publish this article